# Risk Factors for Long-Term Nutritional Disorders One Year After COVID-19: A Post Hoc Analysis of COVID-19 Recovery Study II

**DOI:** 10.3390/nu16234234

**Published:** 2024-12-07

**Authors:** Keiichiro Kawabata, Kensuke Nakamura, Naoki Kanda, Muneaki Hemmi, Shinya Suganuma, Yoko Muto, Arisa Iba, Miyuki Hori, Mariko Hosozawa, Hiroyasu Iso

**Affiliations:** 1Department of Critical Care Medicine, Yokohama City University Hospital, Kanagawa 236-0004, Japan; kawabata.kei.yw@yokohama-cu.ac.jp (K.K.); hemmi.mun.el@yokohama-cu.ac.jp (M.H.); suganuma.shi.en@yokohama-cu.ac.jp (S.S.); 2Department of Emergency and Critical Care Medicine, Hitachi General Hospital, Ibaraki 317-0077, Japan; 3Division of General Internal Medicine, Jichi Medical University, Tochigi 329-0431, Japan; 4Institute for Global Health Policy Research (iGHP), Bureau of International Health Cooperation, National Center for Global Health and Medicine, Tokyo 162-8655, Japanaiba@it.ncgm.go.jp (A.I.); miyhori@it.ncgm.go.jp (M.H.);

**Keywords:** COVID-19, nutritional disorder, long COVID, post-COVID-19 condition, risk factors

## Abstract

**Background/Objectives**: COVID-19 patients develop various clinical symptoms, including malnutrition. However, the risk factors for long-term nutritional disorders remain unclear. Identifying these factors is crucial for preventing nutritional disorders by initiating early nutritional interventions. **Methods**: This was a post hoc analysis of COVID-19 Recovery Study II (CORESII). The study included adult patients hospitalized for COVID-19 and discharged from the hospital. Information, including post-COVID-19 symptoms one month after onset and changes in daily life during the first year, was collected using a self-administered questionnaire sent one year after hospital discharge. We examined the association between baseline characteristics, disease severity, and symptoms that persisted one month after onset with malnutrition disorders one year after onset, defined as a Malnutrition Universal Screening Tool score ≥1, using a logistic regression analysis. **Results:** A total of 1081 patients (mean age of 56.0 years; 34% females; 38% admitted to the intensive care unit) were analyzed. Of these patients, 266 patients (24.6%) had malnutrition one year after onset. In a multivariable logistic regression analysis using variables that were significant in a univariate logistic regression analysis, the following factors were independently associated with malnutrition: BMI < 18.5 kg/m^2^ (odds ratio [95% confidence interval (CI)], 48.9 [14.3–168]), 18.5 ≤ BMI ≤ 20 (10.5 [5.89–18.8]), 30 < BMI (2.64 [1.84–3.75]), length of hospital stay (1.01 [1.00–1.02]), maintenance dialysis (3.19 [1.19–8.61]), and difficulty concentrating one month after onset (1.73 [1.07–2.79]). **Conclusions**: Being underweight or obese, prolonged hospitalization, maintenance dialysis, and difficulty concentrating one month after onset were associated with a risk of malnutrition one year post-illness. Patients with these factors may be at a high risk of long-term nutritional disorders.

## 1. Introduction

Since the outbreak of COVID-19 in December 2019, a number of epidemiological and clinical features have been reported. Even after recovery from acute symptoms, 80% of patients may have long-term symptoms, such as fatigue, pain, coughing, difficulty concentrating, shortness of breath, sleep disturbance, anorexia, taste abnormalities, and psychological distress [1]. The WHO defines long COVID or post-COVID-19 condition (PCC) as the persistence or onset of new symptoms three months after initial SARS-CoV-2 infection, with these symptoms persisting for at least two months without another explanation [2]. Post-affective symptoms, including long COVID-19, have a long-term negative impact on daily life and have become a social issue [3].

Nutritional disorders associated with COVID-19 are another important social issue [4]. The prevalence of malnutrition among hospitalized COVID-19 patients ranges between 37% and 71% [5,6,7], which is higher than that observed in the general hospitalized population [8]. In addition, the prevalence of malnutrition 4–6 months after COVID-19 ranges from 22% to 36%, indicating its persistence in some patients [9,10]. However, few studies have examined the incidence of chronic malnutrition because existing research has focused on the first six months. The risk factors for nutritional disorders and their relationship with PCC have not yet been investigated in detail [11].

Nutritional disorders after COVID-19 may result from various factors. In the acute phase, catabolic stress and metabolic abnormalities due to systemic inflammation [12] and reduced respiratory function [13] are the primary causes of malnutrition. Subsequently, gastrointestinal symptoms related to long COVID, along with olfactory and gustatory disturbances [14], psychiatric symptoms [15], fatigue, muscle weakness [16], and social limitations, may contribute to long-term malnutrition. Malnutrition has various adverse effects, including a higher risk of severe illness [17], prolonged hospitalization [5], an increased risk of reinfection, and increased mortality [18]. The European Society for Clinical Nutrition and Metabolism recommends monitoring the nutritional status of patients recovering from COVID-19, particularly those with PCC, elderly patients, and those with complications. However, this recommendation is not based on specific trials. Acute nutritional screening alone may be insufficient, highlighting the need for additional measures to prevent long-term nutritional disorders [19]. However, nutritional screening and follow-up methods after COVID-19 have not yet been established.

Our study was a post hoc analysis of a prospective cohort study and examined the prevalence and risk factors of nutritional disorders one year post-illness in patients hospitalized for COVID-19. The primary objective was to investigate the need for long-term follow-up and appropriate responses based on individual risk factors, focusing on nutritional disorders following COVID-19. Additionally, this study explored potential associations between nutritional disorders and functional abnormalities during the post-illness phase. Specifically, the present study investigated the prevalence of nutritional disorders one year after COVID-19, the risk factors associated with these disorders, and their relationships with mental health issues and functional abnormalities, such as depression, anxiety, decreased quality of life, fatigue, subjective recovery, and dyspnea.

## 2. Materials and Methods

### 2.1. Study Design and Participants

This was a post hoc analysis of the COVID-19 Recovery Study II (CORESII), a multicenter, prospective study. CORESII investigated PCC, complications, and physical, mental, and social statuses following COVID-19 by tracking patients hospitalized for COVID-19 after their discharge. All procedures involving human participants were reviewed and approved by the institutional review boards of the National Center for Global Health and Medicine (approval number: NCGM-S-004471) and the ethics committees of all participating institutions. Written informed consent was obtained from all participants. This study was supported by MHLW Research on Emerging and Re-emerging Infectious Diseases and Immunization (Program Grant number JPMH21HA2011). The funding source had no role in the research. 

This study was conducted across 20 medical facilities in Japan, including 11 university hospitals and 9 core private hospitals. It involved adult patients hospitalized for COVID-19 and discharged from the hospital between 1 April 2021 and 30 September 2021. This study included patients who were hospitalized for COVID-19 treatment. Patients were excluded if they were deemed ineligible by physicians (e.g., those with dementia or other conditions that could hinder their participation), and the questionnaire was sent to all remaining patients. Follow-up was conducted via a self-administered questionnaire sent one year after discharge to evaluate symptoms, physical function, and mental function post-COVID-19. Since symptoms one month after illness are associated with long COVID-19 [20], the present study assessed post-illness symptoms based on the status of patients at one month.

### 2.2. Data Collection

The study data were collected and managed using REDCap (Research Electronic Data Capture), a secure web-based data capture application hosted at the JCRAC data center of the National Center for Global Health and Medicine [21]. Data collection encompassed several critical areas: patient status (age, sex, height, weight, comorbidities, vaccination history, and the Clinical Frailty Scale score recorded at admission), clinical status (vital signs at admission, results from blood tests, presence of pneumonia on imaging studies, ICU admission, presence of systemic inflammatory response syndrome, length of hospital stay, and incidence of acute respiratory distress syndrome), and treatment (antiviral or steroid therapies, oxygenation strategies, invasive positive pressure ventilation, and extracorporeal membrane oxygenation (ECMO)). The presence of pneumonia on imaging was determined based on findings from chest X-rays or CT scans performed at the discretion of the treating physician within three days before or after admission. Follow-up for this study was conducted using self-administered questionnaires mailed to participants approximately one year after their COVID-19 diagnosis. The questionnaire was developed collaboratively by epidemiologists, public health specialists with medical backgrounds, critical care specialists, and infectious disease specialists. The questionnaire aimed to capture post-illness symptoms, utilizing a predefined list of 26 symptoms derived from the International Severe Acute Respiratory and Emerging Infection Consortium follow-up form [22]. In the present study, 24 symptoms were specifically analyzed, excluding two symptoms (menstrual irregularities and erectile dysfunction) from the original list. The symptoms included fever (>37.5 °C), fatigue, sore throat, rhinorrhea, cough, shortness of breath, chest pain, palpitations, dysgeusia, anosmia, headache, arthralgia, myalgia, muscle weakness, anorexia, vomiting, stomach pain, sleep disorders, difficulty concentrating, brain fog, hair loss, skin rashes, ocular disorders, and dizziness. Each symptom was not specifically defined, and the patients were only asked about its presence at the time of the assessment. Participants were also allowed to report additional symptoms that are not listed but were experienced at the time of the assessment.

In the 1-year follow-up, several assessment tools were utilized to evaluate various aspects of participants’ health and quality of life:EuroQol-5 Dimensions-5 Levels (EQ-5D-5L): The Japanese version of EQ-5D-5L was used to assess quality of life [23]. This tool evaluates five health states: mobility, self-care, usual activities, pain/discomfort, and anxiety/depression, each rated on a five-level scale. Scores for each health state are aggregated to derive the EQ-5D-5L index, with values ranging between 0.025 and 1.000; higher scores indicate better quality of life [24]. An EQ-5D-5L index of <0.8 was used as the cutoff to define a decline in quality of life (QOL).Psychiatric symptoms: Anxiety and depression were assessed using the Hospital Anxiety and Depression Scale (HADS) [25]. This scale evaluates the presence and severity of these psychiatric symptoms. In the assessment of anxiety and depression, we used a cut-off score of ≥8 points for HADS-A and HADS-D. The cut-off for HADS-T was set at ≥16 points [26].Subjective recovery: Participants rated their subjective recovery on a 5-point scale, ranging from 0 (no recovery) to 4 (full recovery). A cut-off for subjective recovery decline was set at ≤1, with a score of 1 indicating “no awareness of recovery”.Shortness of breath: The modified Medical Research Council (mMRC) dyspnea scale was used to evaluate shortness of breath. This scale measures the degree of functional disability caused by dyspnea, with scores ranging from 0 to 4 [27]. A cut-off score of mMRC ≥ 2 was used to define breathlessness, with a score of 2 indicating that “they stop because of shortness of breath while walking at their own pace on a level path” [28].Fatigue: The Eastern Cooperative Oncology Group Performance Status Scale (ECOG PS) was used to evaluate the functional status and level of fatigue experienced by participants [29]. Fatigue was rated on a 5-point scale, from 0 (fully active) to 4 (completely disabled). A cut-off of ECOG PS ≥ 2 was used, indicating that “they cannot work but spend more than 50% of the day out of bed” [30].

Additional information regarding these assessments is provided in Appendix A.

### 2.3. Definitions of Nutritional Disorders

Nutritional disorders in participants were assessed using the Malnutrition Universal Screening Tool (MUST) (Figure 1). This tool evaluates nutritional risk based on three criteria: (1) body mass index (BMI) (1 point for 18.5 ≤ BMI ≤ 20, and 2 points for BMI < 18.5), (2) unplanned weight loss in the past 3 to 6 months (1 point for weight loss between 5% and 10%, and 2 points for weight loss >10%), and (3) no food intake for more than 5 days due to illness. A MUST score ≥ 1 point indicates a medium or higher risk for nutritional disorders [31]. In the present study, nutritional status was assessed using the following three modified items from the MUST score: (1) BMI ≤ 20 one year after illness, (2) weight loss ≥ 5% during the first year after the onset of COVID-19, and (3) anorexia reported one year post-illness. Participants were defined as having a nutritional disorder if they fulfilled at least one of these criteria (Figure 2).

### 2.4. Endpoints

The primary outcome of the present study was nutritional disorders one year after COVID-19. We investigated the relationships between nutritional disorders one year after COVID-19 and various factors, including the patient’s condition at admission and potential risk factors affecting nutritional disorders. Additionally, we examined the relationship between nutritional disorders one year post-COVID-19 and the patient’s status at that time, including QOL, mental health, subjective recovery, dyspnea, and fatigue.

### 2.5. Statistical Analysis

Continuous variables are presented as the mean ± standard deviation or median [first quartile–third quartile], while categorical variables are reported as numbers and percentages. Patients with missing data were excluded from the analysis. To identify independent risk factors for nutritional disorders, both univariable and multivariable logistic regression analyses were performed. The outcome variable was binary, indicating the presence or absence of nutritional disorders one year post-illness. Explanatory variables included age (>75 years) at admission, sex, BMI category at admission (BMI < 18.5, 18.5 ≤ BMI ≤ 20, and BMI > 30), the presence or absence of a medical and hospitalization history (e.g., ICU admission, ventilator use, and treatment), length of hospital stay (days), and the presence or absence of each symptom one month post-illness. Among these explanatory variables, only the length of hospital stay was analyzed as a continuous variable; all other variables were treated as categorical. A univariable logistic regression analysis was conducted for each explanatory variable, and variables showing significant differences were subsequently included in a multivariable logistic regression analysis as covariates. To investigate the relationship between nutritional disorders and mental health issues (anxiety and depression), as well as functional abnormalities (fatigue, QOL, subjective recovery, and dyspnea) one year post-illness, a univariable logistic analysis was performed. The analyses were two-sided and differences between groups were statistically significant if the *p*-value was <0.05. All statistical analyses were performed using R software (version 4.3.1, R Foundation for Statistical Computing, Vienna, Austria).

## 3. Results

### 3.1. Demographic and Clinical Characteristics of the Sample

Between 1 April 2021 and 30 September 2021, 3297 COVID-19 patients were enrolled across the participating centers. Of these patients, 785 were excluded due to ineligibility and, thus, 2512 were included in this study. However, by the 1-year follow-up, 1403 patients had dropped out (1376 did not respond, 13 were unreachable, and 3 had died). A total of 1109 patients completed the self-administered questionnaire one year post-illness. Of these patients, 27 were excluded from the nutritional assessment due to incomplete data on BMI or weight loss. Consequently, 1081 patients were included in the final analysis (Figure 3). The mean age of participants was 55.9 ± 13.5 years, with 718 (66%) being male. The mean BMI was 26.3 ± 5.5 kg/m^2^. Among the 1081 patients, 411 (38.0%) were admitted to the ICU, 342 (31.6%) received invasive positive pressure ventilation, and 41 (3.8%) required treatment with ECMO. The mean length of hospital stay was 15.8 ± 19.9 days.

### 3.2. Primary Outcome

Of the 1081 patients examined, 266 (24.6%) were identified as having nutritional disorders one year after the onset of COVID-19. The mean BMI of all participants at the 1-year follow-up was 26.2 ± 5.2 kg/m^2^, with a 1-year weight variability of 0.48 ± 7.76%. Eighteen patients (1.7%) reported experiencing symptoms of anorexia at 1 year. The univariable logistic regression analysis identified several factors that correlated with nutritional disorders: female sex, BMI < 18.5, 18.5 ≤ BMI ≤ 20, 30 < BMI, dialysis at the time of hospitalization, antiviral therapy during hospitalization, length of hospital stay, oxygen therapy, and standard dose steroid therapy (Table 1). These risks were associated with a MUST score ≥1 one year after COVID-19 onset.

The most prevalent symptoms of COVID-19 sequelae one month after illness were shortness of breath (38.7%), fatigue (34.1%), hair loss (32.4%), and muscle weakness (28.7%). The results of the univariable analysis indicated that 14 out of 24 symptoms assessed after COVID-19 were correlated with nutritional impairment one year later. These symptoms included fever, nasal discharge, coughing, chest pain, palpitations, abnormal sense of smell, headache, muscle weakness, myalgia, anorexia, vomiting, difficulty concentrating, and brain fog (Table 2). A MUST score ≥ 1 one year after COVID-19 onset was associated with the presence of these symptoms one month after illness.

In the multivariable analysis, which included variables that were significant in the univariable analysis (Table 1 and Table 2), the following factors were independently associated with malnutrition: BMI < 18.5 kg/m^2^ (odds ratio [95% confidence interval (CI)], 48.9 [14.3–168]), 18.5 ≤ BMI ≤ 20 (10.5 [5.89–18.8]), 30 < BMI (2.64 [1.84–3.75]), length of hospital stay (1.01 [1.00–1.02]), maintenance dialysis (3.19 [1.19–8.61]), and difficulty concentrating one month after onset (1.73 [1.07–2.79]) (Table 3). The multivariable analysis identified that the factors significantly associated with the MUST score one year after COVID-19 onset were BMI < 18.5, 18.5 ≤ BMI ≤ 20, BMI > 30, length of hospital stay, maintenance dialysis, and difficulty concentrating one month after onset. These factors were independently associated with a MUST score ≥ 1 one year after onset.

### 3.3. Secondary Outcome

The relationships between nutritional disorders and patient status (mental disorders, QOL, subjective recovery, dyspnea, and fatigue) one year after COVID-19 were examined using a univariate analysis. The results showed that nutritional disorders were associated with anxiety (odds ratio [95% CI], 1.89 [1.37–2.62]) and depression (1.54 [1.13–2.10]). In the subjective symptom survey, nutritional disorders were associated with lower QOL (2.19 [1.64–2.94]), lower subjective recovery (1.44 [1.08–1.92]), dyspnea (2.06 [1.47–2.91]), and fatigue (2.39 [1.42–4.01]) (Table 4). Anxiety, depression, reduced quality of life, decreased subjective recovery, breathlessness, and fatigue were related to a MUST score ≥ 1 one year after infection.

## 4. Discussion

The prevalence of nutritional disorders one year post-illness and risk factors at admission were examined in adult patients hospitalized for COVID-19. The prevalence of nutritional disorders one year post-illness was 24.6%. In the present study, 24.6% (approximately a quarter) of patients hospitalized for COVID-19 were found to have nutritional disorders one year after illness. Previous studies reported that the prevalence of malnutrition during the acute phase of COVID-19 was 37–49% [5,6], and 22–36% at 4–6 months post-illness [9,10]. Our study is the first to report on nutritional disability in patients hospitalized for COVID-19 one year after disease onset. Long-term follow-up and tailored responses based on individual risk factors are important and expected to improve patients’ quality of life and support health recovery. Future studies should aim to identify more specific intervention methods and screening tools to prevent and treat nutritional disorders.

The results of the multivariable logistic regression analysis suggested that the following factors were associated with nutritional disorders at one year: low BMI (BMI ≤ 20) at admission, obesity (BMI > 30), a prolonged hospital stay, maintenance dialysis, and difficulty concentrating one month after illness. Nutritional deficiencies at one year were associated with anxiety, depression, decreased QOL, lower subjective recovery, dyspnea, and fatigue. Difficulty concentrating is a symptom of neuropathy after COVID-19; therefore, nutritional disorders might possibly result from neurologic symptoms. Neurological symptoms are experienced by 36–57% of patients hospitalized for COVID-19 [32]. Anorexia is a common neurological symptom with a prevalence of 24–31% in hospitalized COVID-19 patients and is often a social issue because it is prolonged [33,34]. In the present study, anorexia was also associated with nutritional disorders in univariate analyses. Possible mechanisms for neurological symptoms include angiotensin-converting enzyme 2 (ACE-2) receptor involvement, hypoxic metabolic disturbances, nervous system inflammation, blood–brain barrier damage, and neurotransmitter abnormalities [35,36]. In particular, with respect to difficulty concentrating, SARS-CoV-2 binds its spike proteins to ACE-2 receptors and invades host cells, leading to various symptoms [37,38]. A potential cause is the increased vulnerability of the frontal and cingulate cortices (areas involved in cognitive functions, such as concentration and motivation), where ACE-2 receptors are highly expressed [39,40,41].

Long-term nutritional disorders after COVID-19 may be caused by several factors, including decreased nutrient intake due to anorexia nervosa [42], accelerated proteolysis resulting from elevated levels of various inflammatory cytokines and tumor necrosis factors [43], increased resting energy expenditure [44], and decreased nutrient intake and malabsorption associated with gastrointestinal symptoms among post-infection sequelae [45]. Although it is challenging to identify risk factors for all symptoms because of the wide variety of post-affective symptoms [7], the results of the multivariate analysis revealed that other than difficulty concentrating, the remaining 23 symptoms examined (particularly gastrointestinal symptoms, such as abdominal pain and vomiting, respiratory symptoms, circulatory symptoms, muscle weakness, joint pain, and myalgia) were not significantly associated with nutritional impairment at one year. The widespread distribution of ACE-2 receptors in the gastrointestinal tract means that SARS-CoV-2 can cause gastrointestinal symptoms [46]; however, its effects on nutritional impairment may be limited because gastrointestinal symptoms generally do not persist as long as neurological symptoms [47].

The results of this study showed that patients with nutritional disorders after COVID-19 were more likely to exhibit a combination of psychiatric symptoms (such as anxiety and depression) and functional impairments (including decreased quality of life, lower subjective recovery, dyspnea, and fatigue). These psychiatric symptoms and functional impairments negatively affect daily life, leading to a reduced quality of life, and nutritional disorders may contribute to a detrimental cycle after the disease [48]. However, the relationship regarding cause and effect remains unclear because mental impairment, functional impairment, decreased QOL, and nutritional impairment may interact with and influence one another. Although improvements in nutritional status have been reported with several nutritional therapies [49,50], these effects have been limited, and there have still been no interventions that have achieved long-term improvements in nutritional outcomes. Since the present results indicate that the causes of nutritional disorders after COVID-19 disease are complex, comprehensive care may be needed in addition to nutritional therapy.

There was no relationship between ICU admission or severity of illness in the acute phase (severe ARDS, invasive positive pressure ventilation, or treatment with ECMO) and malnutrition one year post-illness. In several prior studies, ICU admission and the severity of COVID-19 have been identified as risk factors for malnutrition; however, few studies have conducted nutritional assessments beyond six months post-illness [51,52]. Some ICU survivors develop ICU-AW, which is associated with persistent weakness and fatigue lasting months after discharge, complicated by physical and neuropsychiatric disability and impaired quality of life. Severe illness, not limited to COVID-19, may have a negative impact on the nutritional status [53]. Post-intensive care syndrome associated with COVID-19 may have a distinct impact on nutritional status because cognitive impairment is often more pronounced than physical impairment [54]. While previous studies showed that ICU admission and the severity of illness could affect nutritional status, the present results did not find a significant relationship between acute-phase severity and nutritional impairment at one year. In this study, nutritional disorders were defined based on a MUST score ≥ 1; therefore, the prevalence of mild to severe nutritional disorders was examined. If limited to severe nutritional disorders, ICU admission may have been a risk factor, because severe nutritional disorders are more common in patients admitted to the ICU than in patients admitted to general wards.

Among complications, maintenance dialysis has been identified as a risk factor for nutritional disorders. Dialysis patients were the prominent group, not only in terms of mortality, functional impairment, and acute nutritional disorders caused by COVID-19 but also with regard to long-term nutritional disorders at one year [55]. These patients are prone to impaired catabolic reactions due to COVID-19, often characterized by weight loss and muscle wasting [55]. However, nutritional disorders in dialysis patients are not limited to COVID-19, as this patient population faces numerous factors contributing to malnutrition, even in the absence of infection [56]. Dialysis patients require intensified and comprehensive nutritional interventions to address their heightened risks effectively.

## 5. Limitations

Firstly, the patients’ nutritional status prior to COVID-19 infection was not assessed. Consequently, it is possible that nutritional impairments were already present before infection. Secondly, data on food intake were not collected, leaving the impact of dietary changes on body weight unclear.

In the present study, being underweight or obese was identified as a risk factor for nutritional dysfunction 1 year after COVID-19; however, the inclusion of BMI as a criterion for nutritional dysfunction may have strongly affected the results of the analysis due to the study design. However, being underweight or obese is a known risk factor for nutritional disorders in the acute phase of COVID-19 recovery [57], and the potential for long-term adverse effects on nutritional status post-disease through inflammation-related metabolic disturbances and long COVID-19 has been reported [58]. These findings align with the results of our current study, which also highlights the risk of long-term nutritional issues in these populations.

## 6. Conclusions

The present results highlight the potential for prolonged nutritional disability in some patients following COVID-19 and emphasize the importance of long-term nutritional follow-up for those hospitalized with the disease. Being underweight or obese, prolonged hospitalization, dialysis, and difficulty concentrating at 1 month were associated with nutritional disability after one year of illness. Patients with these factors are at high risk for persistent nutritional disorders and require a greater need for ongoing monitoring and support to address their nutritional needs effectively.

## Figures and Tables

**Figure 1 nutrients-16-04234-f001:**
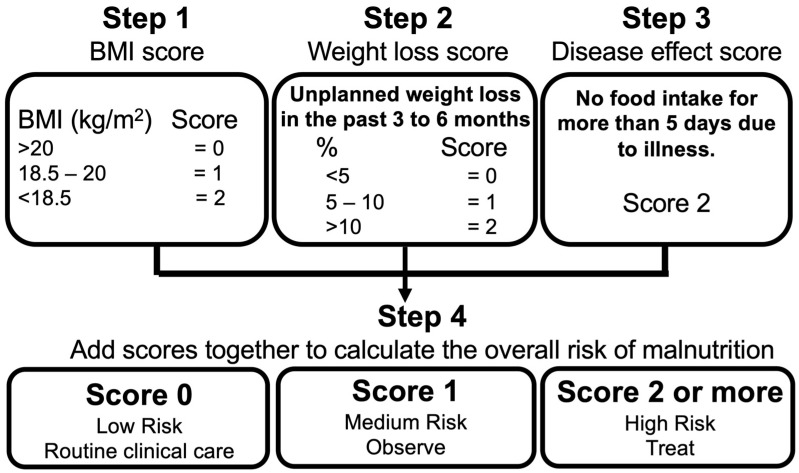
Malnutrition Universal Screening Tool (MUST).

**Figure 2 nutrients-16-04234-f002:**
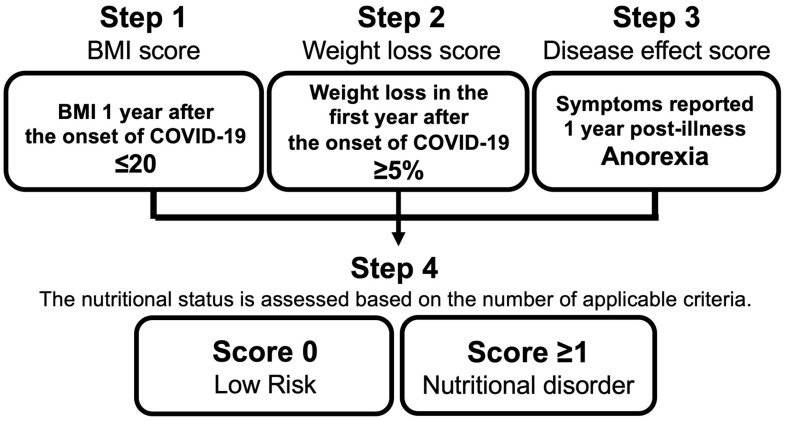
Nutritional assessment tool modified from the MUST score.

**Figure 3 nutrients-16-04234-f003:**
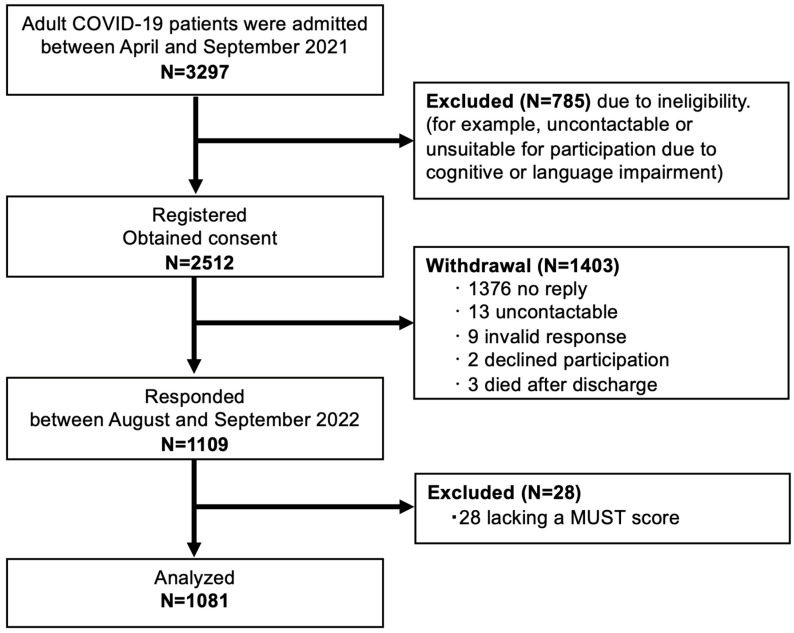
Flow chart of the study population.

**Table 1 nutrients-16-04234-t001:** Patient characteristics according to MUST score.

Group	Overall	MUST = 0	MUST ≥ 1	*p*
n	1081	815	266	value
Female	363 (34)	237 (29.1)	126 (47.4)	0.009 *
Age > 75 years	89 (8.2)	62 (7.7)	27 (10.3)	0.188
BMI < 18.5	33 (3.0)	3 (0.4)	30 (11.3)	<0.001 *
18.5 ≤ BMI ≤ 20.0	66 (6.1)	21 (2.6)	45 (17.0)	<0.001 *
30.0 < BMI	229 (21.2)	155 (19.0)	74 (27.8)	0.002 *
Clinical frailty scale ^†^ > 4	53 (4.9)	35 (4.3)	18 (6.7)	0.424
Vaccinated	121 (11.2)	90 (11.3)	31 (11.8)	0.807
SIRS on admission	432 (40.0)	331 (43.9)	101 (40.4)	0.333
Pneumonia images ^‡^	977 (90.3)	740 (90.8)	233 (87.6)	0.131
Comorbidities				
None	385 (35.6)	289 (35.5)	96 (36.1)	0.852
Cancer	63 (5.8)	47 (5.8)	16 (6.0)	0.881
Myocardial infarction	22 (2.0)	17 (2.1)	5 (1.9)	0.831
Heart failure	20 (1.8)	12 (1.5)	8 (3.0)	0.107
Paralysis	5 (0.5)	4 (0.5)	1 (0.4)	0.811
Cerebrovascular disorder	40 (3.7)	30 (3.7)	10 (3.8)	0.953
Peripheral vascular disease	9 (0.8)	5 (0.6)	4 (1.5)	0.166
Hypertension	359 (33.2)	273 (33.5)	86 (32.3)	0.726
DM	242 (22.4)	185 (22.7)	57 (21.4)	0.666
Dyslipidemia	199 (18.4)	151 (18.5)	48 (18.0)	0.860
Interstitial lung disease	6 (0.6)	3 (0.4)	3 (1.1)	0.148
COPD	27(2.5)	22 (2.7)	5 (1.9)	0.457
Asthma	76 (7.0)	52 (6.4)	24 (9.0)	0.144
Peptic ulcer	9 (0.8)	9 (1.1)	0 (0)	0.085
Chronic liver disease	61 (5.6)	43 (5.3)	18 (6.8)	0.361
CKD	45 (4.2)	30 (3.7)	15 (5.6)	0.165
Dialysis patient	20 (1.8)	11 (1.4)	9 (3.4)	0.033 *
Collagen disease	34 (3.1)	25 (3.1)	9 (3.4)	0.798
HIV/AIDS	8 (0.7)	7 (0.9)	1 (0.4)	0.425
On immunosuppressive drugs	24 (2.2)	19 (2.3)	5 (1.9)	0.665
Psychiatric disorder	46 (4.3)	31 (3.8)	15 (5.6)	0.198
Severe ARDS	71 (6.6)	47 (5.8)	24 (9.0)	0.062
Antiviral drug therapy	795 (74.4)	613 (75.2)	182 (68.4)	0.029 *
Length of stay, mean (SD), days	15.8 (19.9)	14.8 (16.1)	19.2 (28.6)	0.002 *
ICU admission	411 (38.1)	303 (37.2)	109 (41.0)	0.268
Oxygen therapy	852 (78.7)	660 (81.0)	192 (72.2)	0.002 *
Mechanical ventilation	342 (31.6)	251 (30.8)	91 (34.2)	0.299
ECMO	41 (3.8)	27 (3.3)	14 (5.3)	0.148
Steroid treatment (standard dose)	669 (61.8)	521 (63.9)	148 (55.6)	0.016 *
Steroid treatment (pulse dose)	232 (21.4)	172 (21.1)	60 (22.6)	0.617

Variables were expressed as numbers with percentages unless otherwise indicated, and all variables were compared using univariate logistic regression analysis. * *p*-values < 0.05 indicate a significant difference. ^†^ Clinical frailty scale ranges from 1 (very fit) to 9 (terminally ill). ^‡^ Pneumonia images are diagnosed on X-rays or CT within 3 days of admission. Abbreviations: MUST—malnutrition universal screening tool score; BMI—body mass index; SIRS—systemic inflammatory response syndrome; COPD—chronic obstructive pulmonary disease; HIV/AIDS—human immunodeficiency virus/acquired immunodeficiency syndrome; ARDS—Acute respiratory distress syndrome; SD—standard deviation; ICU—intensive care unit; ECMO—extracorporeal membrane oxygenation.

**Table 2 nutrients-16-04234-t002:** Survey of COVID-19 symptoms at 1 month of onset according to MUST score.

Group	Overall	MUST = 0	MUST ≥ 1	*p*
n	1081	815	266	value
Fever (>37.5 °C)	69 (6.3)	43 (5.3)	26 (9.7)	0.009 *
Fatigue	369 (34.1)	272 (33.4)	97 (36.3)	0.356
Sore throat	68 (6.3)	49 (6.0)	19 (7.1)	0.510
Rhinorrhea	67 (6.2)	40 (4.9)	27 (10.1)	0.002 *
Cough	220 (20.3)	153 (18.8)	67 (25.1)	0.024 *
Shortness of breath	419 (38.7)	309 (38.0)	110 (41.2)	0.318
Chest pain	92 (8.5)	59 (7.2)	33 (12.4)	0.009 *
Palpitation	125 (11.6)	78 (9.6)	47 (17.6)	<0.001 *
Dysgeusia	178 (16.5)	126 (15.5)	52 (19.5)	0.119
Anosmia	154 (14.2)	103 (12.6)	51 (19.1)	0.012 *
Headache	92 (8.5)	61 (7.5)	31 (11.6)	0.034 *
Arthralgia	103 (9.5)	71 (8.7)	32 (12.0)	0.11
Myalgia	106 (9.8)	69 (8.5)	37 (13.9)	0.010 *
Muscle weak	310 (28.7)	216 (26.5)	94 (35.2)	0.006 *
Anorexia	86 (8.0)	50 (6.1)	36 (13.5)	<0.001 *
Vomiting	36 (3.3)	22 (2.7)	14 (5.2)	0.043 *
Stomach pain	19 (1.8)	11 (1.3)	8 (3.0)	0.074
Sleep disorder	218 (20.2)	159 (19.5)	59 (22.1)	0.346
Difficulty concentrating	221 (20.4)	148 (18.2)	73 (27.3)	0.001 *
Blain fog	156 (14.4)	107 (13.1)	49 (18.4)	0.033 *
Hair loss	351 (32.4)	249 (30.6)	102 (38.2)	0.018 *
Skin rashes	83 (7.7)	59 (7.2)	24 (9.0)	0.343
Ocular disorders	126 (11.7)	87 (10.7)	39 (14.6)	0.079
Dizziness	94 (8.7)	64 (7.9)	30 (11.2)	0.085

Variables were expressed as numbers with percentages and compared using univariate logistic regression analysis. * *p*-values < 0.05 indicate a significant difference. Abbreviations: MUST—malnutrition universal screening tool score.

**Table 3 nutrients-16-04234-t003:** Odds ratios (95% confidence intervals) for nutritional disorders one year after onset by patient characteristics.

Group	Univariate Analysis	Multivariate Analysis	
	Odds Ratio (95% CI)	Odds Ratio (95% CI)	*p*-Value
Female	2.19 (1.65–2.92)	1.41 (0.99–2.01)	0.055
BMI < 18.5	34.4 (10.4–114)	48.9 (14.3–168)	<0.001 *
18.5 ≤ BMI ≤ 20.0	7.70 (4.49–13.2)	10.5 (5.89–18.8)	<0.001 *
30.0 < BMI	1.64 (1.19–2.26)	2.62 (1.84–3.75)	<0.001 *
Dialysis patient	1.23 (1.02–1.49)	3.19 (1.19–8.61)	0.021 *
Antiviral therapy	0.71 (0.53–0.97)	0.95 (0.65–1.40)	0.806
Length of stay, days	1.01 (1.00–1.02)	1.01 (1.00–1.02)	0.003 *
Oxygen therapy	0.61 (0.44–0.84)	0.84 (0.51–1.36)	0.742
Steroid treatment (standard dose)	0.71 (0.53–0.94)	0.78 (0.54–1.14)	0.202
COVID-19 symptoms at one month			
Fever (>37.5 °C)	1.94 (1.17–3.23)	1.11 (0.59–2.11)	0.742
Rhinorrhea	2.19 (1.32–3.64)	1.72 (0.90–3.31)	0.102
Cough	1.46 (1.05–2.02)	0.90 (0.59–1.37)	0.623
Chest pain	1.81 (1.16–2.85)	1.18 (0.66–2.11)	0.570
Palpitation	2.03 (1.37–3.00)	1.32 (0.79–2.19)	0.285
Anosmia	1.60 (1.10–2.32)	1.02 (0.65–1.62)	0.922
Headache	1.63 (1.03–2.57)	1.25 (0.69–2.26)	0.461
Myalgia	1.75 (1.14–2.67)	0.99 (0.57–1.75)	0.982
Muscle weakness	1.52 (1.13–2.04)	1.20 (0.80–1.80)	0.372
Anorexia	2.39 (1.52–3.77)	1.43 (0.80–2.58)	0.223
Vomiting	2.00 (1.01–3.97)	0.62 (0.24–1.60)	0.326
Difficulty concentrating	1.70 (1.23–2.35)	1.73 (1.07–2.79)	0.025 *
Brain fog	1.49 (1.03–2.16)	0.75 (0.43–1.30)	0.309
Hair loss	1.41 (1.06–1.89)	0.99 (0.69–1.42)	0.937

A Multivariable logistic regression analysis was performed using variables that were statistically significant in the univariate logistic regression analysis (Table 1, Table 2 and Table 3). * *p*-values < 0.05 indicate a significant difference. Abbreviations: CI—confidence interval; BMI—body mass index.

**Table 4 nutrients-16-04234-t004:** Odds ratios (95% confidence intervals) for nutritional disorders one year after onset and physical and mental health status.

Group	MUST = 0	MUST ≥ 1	Univariate Analysis	
N	815	266	Odds Ratio (95%CI)	*p*-Value
HADS Anxiety ≥ 8	144 (18.2)	75 (29.6)	1.89 (1.37–2.62)	<0.001 *
HADS Depression ≥ 8	181 (22.8)	81 (31.3)	1.54 (1.13–2.10)	0.006 *
HADS Total ≥ 16	132 (16.9)	76 (30.3)	2.13 (1.53–2.95)	<0.001 *
EQ-5D-5L < 0.80	206 (25.7)	112 (43.1)	2.19 (1.64–2.94)	<0.001 *
Recovery awareness ^†^ ≤ 1	248 (30.5)	103 (38.7)	1.44 (1.08–1.92)	0.013 *
Breathlessness score ^‡^ ≥ 2	116 (14.7)	66 (26.2)	2.06 (1.47–2.91)	<0.001 *
Fatigue score ^§^ ≥ 2	37 (4.6)	27 (10.3)	2.39 (1.42–4.01)	<0.001 *

Categorical variables were expressed as numbers with percentages and compared using univariate logistic regression analysis. * *p*-values < 0.05 indicate a significant difference. ^†^ Recovery awareness score of 1 indicates ‘no awareness of recovery’. ^‡^ A breathlessness score of 2 indicates that they stop due to shortness of breath while walking at their own pace on flat paths. ^§^ A fatigue score of 2 indicates that they cannot work but spend more than 50% of the day out of bed. Abbreviations: HADS—hospital anxiety and depression scale; EQ-5D-5L—EuroQol-5 dimensions-5 levels.

## Data Availability

All patients consented to publication. The data that support the findings of this study are available on reasonable request to the CORESII Research office (cores2@it.ncgm.go.jp).

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
