# Peer review of "Risk Factors for Long-Term Nutritional Disorders One Year After COVID-19: A Post Hoc Analysis of COVID-19 Recovery Study II"

_nutrients, 2024, doi:10.3390/nu16234234_

Round 1

Reviewer 1 Report

Comments and Suggestions for Authors

In this paper, the authors describe the risk factors for long-term nutritional disorders following COVID-19. The study is interesting and scientifically sound, with clearly presented results. However, the following issues need to be addressed:

In the conclusion, the authors wrote: "Being underweight, being obese, a long-term stay at hospital, receiving dialysis at 30 hospitalization, and concentration loss 1 month after onset were associated with a risk of malnutrition 1 year post-illness."

The term "concentration loss" requires clarification. Would terms like attention deficit, difficulty concentrating, or cognitive impairment be more appropriate? Please revise this sentence for clarity, as this term is repeated multiple times in the manuscript.

The phrase "discharged alive" is used repeatedly in the manuscript, such as in the sentence: "It involved adult patients who had been hospitalized for COVID-19 and were discharged alive between April 1, 2021, and September 30, 2021." Please change the term throughout the manuscript and use another, more appropriate term.

The sentence "Concentration less is a symptom of neuropathy after COVID-19, and anorexia nervosa may be one of the causes of nutritional impairment" should be corrected.

The sentence "Factors associated with nutritional disability after 1 year of illness were underweight, obesity, prolonged hospitalization, dialysis, and concentration loss at 1 month" should also be revised.

To enhance the readability of the manuscript, add a brief conclusion after each results section to highlight the most important and relevant findings from each table. 

Comments on the Quality of English Language

Minor editing is required to improve the clarity of some terms used in the manuscript.

Author Response

Point 1: In the conclusion, the authors wrote: "Being underweight, being obese, a long-term stay at hospital, receiving dialysis at 30 hospitalization, and concentration loss 1 month after onset were associated with a risk of malnutrition 1 year post-illness."

The term "concentration loss" requires clarification. Would terms like attention deficit, difficulty concentrating, or cognitive impairment be more appropriate? Please revise this sentence for clarity, as this term is repeated multiple times in the manuscript.:

Our response:

We thank the reviewer’s comment. We revised "concentration loss" to "difficulty concentrating" throughout the manuscript. We added the explanation in method section as follows.

Lines: 128-129

"Each symptom was not specifically defined, and the patients were only asked about its presence at that time of the assessment.”

Point 2: The phrase "discharged alive" is used repeatedly in the manuscript, such as in the sentence: "It involved adult patients who had been hospitalized for COVID-19 and were discharged alive between April 1, 2021, and September 30, 2021." Please change the term throughout the manuscript and use another, more appropriate term.

Our response: We thank the reviewer for this helpful comment. We revised "discharged alive" to "discharged from the hospital" throughout the manuscript.

Lines: 93-95

“It involved adult patients who had been hospitalized for COVID-19 and were discharged from the hospitalbetween April 1, 2021 and September 30, 2021.”

Point 3: The sentence "Concentration less is a symptom of neuropathy after COVID-19, and anorexia nervosa may be one of the causes of nutritional impairment" should be corrected.

Our response: We thank the reviewer for this meaningful comments. We have made the following revisions.

Lines: 318-323

“Difficulty concentrating is a symptom of neuropathy after COVID-19, therefore, nutritional disorders might possibly result from neurologic symptoms. Neurological symptoms are experienced by 36-57% of patients hospitalized for COVID-19.[32] Anorexia is a common neurological symptom with a prevalence of 24-31% of COVID-19 hospitalized patients and is often a social issue because it is prolonged.[33, 34] In the present study, anorexia was also associated with nutritional disorders in univariate analyses.”

Point 4: The sentence "Factors associated with nutritional disability after 1 year of illness were underweight, obesity, prolonged hospitalization, dialysis, and concentration loss at 1 month" should also be revised.

Our response: We thank the reviewer for this important comment. We have revised the sentence in the Conclusions as follows.

Lines: 405-407

“Being underweight, obesity, prolonged hospitalization, dialysis, and difficulty concentrating at 1 month were associated with nutritional disability after one year of illness.”

Point 5: To enhance the readability of the manuscript, add a brief conclusion after each results section to highlight the most important and relevant findings from each table.

Our response: We appreciated the reviewer’s comment on this point. We have added highlights at the end of each section to emphasize the key results.

Lines: 227-228

“ These risks were associated with a MUST score ≥1 at one year after COVID-19 onset. ”

Lines: 247-248

“A MUST score ≥1 one year after COVID-19 onset was associated with the presence of these symptoms at one month after illness.”

Lines: 263-264

“ These factors were independently associated with a MUST score ≥1 one year after onset.”

Lines: 278-280

“ Anxiety, depression, reduced quality of life, decreased subjective recovery, breathlessness, and fatigue were related to a MUST score ≥1 one year after infection.”

Reviewer 2 Report

Comments and Suggestions for Authors

In the current study the authors investigated the prevalence of nutritional disorders one year after COVID-19, risk factors associated with  the development of these disorders, and the relationships between nutritional disorders 78 and mental health issues as well as functional abnormalities, such as depression, anxiety, decreased quality of life, fatigue, subjective recovery and dyspnea. 

Some suggestions:

1. The criteria for excluding patients must be clearly defined. At lines 96-98 you wrote: “Patients were excluded if they were deemed ineligible by physicians (e.g., those with dementia or other conditions that may hinder their cooperation in the study)”. This are all the exclusion criteria?

2. line 93 - Give please details concerning the “medical facilities in Japan”?

2. Add please the questionnaire as a supplementary file. Also please specify which is the level of professional training of the person who drew up the questionnaire.

3. In the abstract you wrote that you followed up the symptoms one month after and one year after COVID-19 and at line 101 that you followed up the symptoms one month after. See please also line 116. Please clarify.

4. In my opinion Supplementary Figure 1 must be added in the article.

5. You forgot to add Figures 1 and 2 in the article.

6. In my opinion, two major problems of the study are represented by the fact that the baseline nutritional status before COVID-19 infection is not assessed and also that the food intake was not recorded.

Author Response

Point 1: The criteria for excluding patients must be clearly defined. At lines 96-98 you wrote: “Patients were excluded if they were deemed ineligible by physicians (e.g., those with dementia or other conditions that may hinder their cooperation in the study)”. This are all the exclusion criteria?

Our response: Thank you for your comment. This is all the exclusion criteria. We have added the following supplementary information.

Lines: 95-98

 “Patients were excluded if they were deemed ineligible by physicians (e.g., those with dementia or other conditions that could hinder their participation) , and the questionnaire was sent to all remaining patients.

Point 2: line 93 - Give please details concerning the “medical facilities in Japan”?

Our response: We appreciated the reviewer’s comment on this point. We have added the details as below.

Lines: 92-93

“This study was conducted across 20 medical facilities in Japan, including 11 university hospitals and 9 core private hospitals.”

Point 3: Add please the questionnaire as a supplementary file. Also please specify which is the level of professional training of the person who drew up the questionnaire.

Our response: We thank the reviewer for this comment. We regret to inform you that the CORESII study is funded and managed by the Ministry of Health, Labour and Welfare, and therefore, public disclosure is not permitted. We have added the following additional information.

Lines: 117-119

”The questionnaire was developed collaboratively by epidemiologists, public health specialists with medical backgrounds, critical care specialists, and infectious disease specialists.”

The following text has been included in the manuscript as the Data Availability Statement.

Lines: 428-430

“The data that support the findings of this study are available on reasonable request to the CORESII Research office (cores2@it.ncgm.go.jp).”

Point 4: In the abstract you wrote that you followed up the symptoms one month after and one year after COVID-19 and at line 101 that you followed up the symptoms one month after. See please also line 116. Please clarify.

Our response: We thank the reviewer for this meaningful comment. I have revised the following sentence in the abstract to the one below.

Lines: 19-22

”Information, including post-COVID-19 symptoms one month after onset and changes in daily life during the first year, was collected using a self-administered questionnaire sent one year after hospital discharge.”

Point 5: In my opinion Supplementary Figure 1 must be added in the article.

Our response: We thank the reviewer for this comment. We have added Supplementary Figure 1 as the new Figure 1 in the article.

Point 6: You forgot to add Figures 1 and 2 in the article.

Our response: We thank you for your guidance. We have added Figures 1 and 2 as the new Figure 2 and 3 in the article.

Point7: In my opinion, two major problems of the study are represented by the fact that the baseline nutritional status before COVID-19 infection is not assessed and also that the food intake was not recorded.

Our response: We wish to thank the reviewer for this comment. We agree that these points. I have revised the beginning of the Limitations section to emphasize these two major issues. We have revised the Limitations section as follows.

Lines: 389-401

“Firstly, the nutritional status prior to COVID-19 infection was not assessed. Con-sequently, it is possible that nutritional impairments were already present before infec-tion. Secondly, data on food intake were not collected, leaving the impact of dietary changes on body weight unclear.”